# Comparison of Motor Scores between OFF and ON States in Tremor-Dominant Parkinson’s Disease after MRgFUS Treatment

**DOI:** 10.3390/jcm11154502

**Published:** 2022-08-02

**Authors:** Chunyu Yin, Rui Zong, Ge Song, Jiayou Zhou, Longsheng Pan, Xuemei Li

**Affiliations:** 1Department of Cadres’ Outpatient, the First Medical Center, Chinese PLA General Hospital, Beijing 100853, China; yincy301@163.com; 2Department of Neurosurgery, the First Medical Center, Chinese PLA General Hospital, Beijing 100853, China; zongrui11@126.com (R.Z.); zhoujiayou301@163.com (J.Z.); 3Department of Health Care, 305 Hospital of Chinese PLA, Beijing 100017, China; sganthem@163.com

**Keywords:** MRgFUS, TDPD, MDS-UPDRS, CRST, LEDD

## Abstract

Objective: To compare the motor function improvements in ON and OFF states in tremor-dominant Parkinson’s disease (TDPD) patients within one year of follow-up after ablation of the unilateral ventral intermediate nucleus of the thalamus (Vim) by magnetic resonance imaging-guided focused ultrasound surgery (MRgFUS). Methods: A total of nine consecutive patients confirmed with TDPD who underwent unilateral Vim ablation by MRgFUS between April 2019 and September 2019 were included. The Movement Disorder Society Unified Parkinson’s Disease Rating Scale part III (MDS-UPDRSIII) and Clinical Rating Scale for Tremor (CRST) were performed in the ON and OFF stages to distinguish the surgical effects from drug therapy effects. The adverse events and adjustment of drug doses were also recorded. Results: The preoperative MDS-UPDRSIII score in OFF and ON states was 55.0 (48.0, 65.5) and 26.0 (17.0, 27.0), while the CRST score was 46.0 (39.5, 53.5) and 20.0 (13.0, 23.5), respectively; the Levodopa equivalent dose was 600 (456, 600) mg/d. At 1 year after operation, the total MDS-UPDRSIII score and CRST score were 40.0 (30.0, 60.5) and 16.0 (10.0, 29.5) in the OFF state, and 21.0 (17.5, 27.0) and 2.0 (1.5, 7.0) in the ON state, respectively. Compared with the preoperative levels, follow-up at the two-time points (three months and one year after operation) showed the total MDS-UPDRSIII score, as well as MDS-UPDRSIII tremor, bradykinesia, and rigidity scores of contralateral limbs all significantly improved in OFF state. However, in the ON state, only the total MDS-UPDRSIII score and tremor score of contralateral limbs significantly improved. The total CRST score and the CRST (A + B) score of contralateral limbs significantly improved at three months and one year after the operation compared with before the operation in both ON and OFF states. The Levodopa equivalent dose at one and three months were not significantly different from the preoperative dose (*p* > 0.05). No serious adverse responses were observed. Conclusion: Treating TDPD with unilateral Vim ablation by MRgFUS could improve the symptoms of limb tremor and the other core symptoms, such as bradykinesia and rigidity, as well as some non-motor symptoms and the symptoms of ipsilateral limbs.

## 1. Background

Parkinson’s disease (PD) is a common neurodegenerative disease whose clinical manifestations include motor symptoms (tremor, bradykinesia, and rigidity) and non-motor symptoms (cognitive disorder and mental disorders). The severity of motor and non-motor symptoms in PD varies substantially, and the age of disease onset and involvement of limbs can also vary, thus making PD a disease with certain heterogeneity. Previous studies using clustering analysis have further classified PD into different subtypes, such as early-onset PD, tremor-dominant PD (TDPD), non-tremor dominant PD accompanied with cognitive impairment and depression, and rapid progression PD not accompanied by cognitive impairment [1]. TDPD is described with various classification criteria, which can be classified by the ratio of mean tremor score to postural instability/gait difficulty (PIGD) score ≥ 1.5 on Unified Parkinson’s Disease Rating Scale (UPDRS), or the ratio ≥ 1.15 in the Movement Disorder Society Unified Parkinson’s Disease Rating Scale (MDS-UPDRS) [2].

The characteristics of TDPD include the younger age of disease onset and comparatively slow disease progression. The initial and dominant symptom is usually the static tremor of unilateral limbs, while rigidity and bradykinesia are usually not evident [3]. Regarding the drug therapy efficacy, it is generally considered that anticholinergic agents (ACA) have the highest treatment efficacy for TDPD, followed by dopaminergic receptor agonists (DRA) and levodopa (L-DOPA). However, ACA is only suitable for relatively young TDPD patients without cognitive impairments. DRA has moderate treatment effects on tremors, but it is generally difficult to meet the treatment requirements of TDPD. L-DOPA shows poor tremor-inhibiting effects with various side effects when treating TDPD patients, and can induce motor fluctuations or dyskinesia at an early stage [4,5]. In summary, the efficacy of medical therapy for TDPD is suboptimal.

Surgical treatments, including deep brain stimulation (DBS) and radiofrequency ablation, have also been used to treat PD patients. Magnetic resonance imaging-guided focused ultrasound surgery (MRgFUS), which has recently emerged as a treatment option for PD patients with suboptimal responses to drug therapy, has relatively minimal invasiveness and high safety. In 2018, the FDA approved MRgFUS for the treatment of TDPD, and the Chinese National Medical Products Administration (NMPA) recently approved it for the treatment of ET and TDPD [6]. However, the long-term efficacy of MRgFUS in treating TDPD, especially that hidden under pharmaceutical adjuvant therapy (i.e., OFF state), has been rarely reported. Therefore, in this study, we prospectively followed up the patients for one year and investigated the motor function, adverse reactions, and adjustment of drug doses in ON and OFF states in TDPD patients after the unilateral ventral intermediate nucleus of thalamus (Vim) ablation by MRgFUS.

## 2. Materials and Methods

This prospective, single-arm clinical study was approved by the Ethics Committee of the General Hospital of the People’s Liberation Army. The Department of Neurosurgery and Radiology of the First Medical Center and the Neurology Section of the Outpatient Cadre Diagnosis and Treatment Department of the hospital participated in this study (Ethics approval No.: 2018lunshendi021 and 2018lunshendi021-1). All of the participants signed informed consent.

### 2.1. Inclusion and Exclusion Criteria

The inclusion criteria were as follows: (1) patients diagnosed with primary PD, and clinical classification was TDPD; (2) patients who underwent drug therapy before the operation; however, the management of symptoms was unsatisfactory, or the patient was unable to tolerate the side effects of drug therapy; and (3) those who consented to undergo unilateral Vim ablation by MRgFUS.

The exclusion criteria were patients: (1) accompanied with other intracranial diseases, such as a tumor, arterial aneurysm, or vascular malformation; (2) accompanied with cognitive impairment, i.e., Mini-mental State Examination (MMSE) score ≤ 24; (3) with a history of psychological diseases; (4) who underwent DBS surgery or other stereotactic ablation before; (5) with contraindications to MRI; (6) in poor physical condition and unable to tolerate the operation, such as cardiac insufficiency or coagulation disorders; and (7) with a skull density ratio (SDR) < 0.3.

### 2.2. Disease Condition Assessment

Each patient was assessed by at least two neurological physicians, according to the latest version Movement Disorder Society Unified Parkinson’s Disease Rating Scale (MDS-UPDRS). Although the scores were assessed with at least two neurological physicians, the evaluation (both ON/OFF state and contralateral/ipsilateral side) was not blind or randomized. Therefore, the criteria for diagnosing TDPD included the ratio of average tremor score (i.e., the average score of questions 2.10, 3.15a, 3.15b, 3.16a, 3.16b, 3.17a, 3.17b, 3.17c, 3.17d, 3.17e, and 3.18) to average PIGD score (i.e., the average score of questions 2.12, 2.13, 3.10, 3.11, and 3.12) ≥ 1.15. After the baseline clinical assessment and imaging examinations, motor examinations in ON and OFF states were performed at three months and one year after the operation, using MDS-UPDRSIII and Clinical Rating Scale for Tremor (CRST). The adverse events and Levodopa equivalent dose were also recorded.

### 2.3. Surgical Treatment

Experienced neurosurgeons performed all operations. The MRgFUS equipment (650 kHz Exablate Neuro; InSightec, Haifa, Israel) was used for the operation. The side of surgical ablation was selected according to the severity of limb symptoms and personal willingness. The target was Vim. The core temperature of at least 54–56 °C, which was used twice, and at least 57–60 °C used once, represented the ablation endpoint; the tremor in patients was well inhibited.

### 2.4. Statistical Analysis

SPSS version 22.0 software (IBM, Armonk, NY, USA) was used for the statistical analysis. Continuous variables with normal distribution were described by means and standard deviations, and those with non-normal distribution by median (P25, P75). Comparison of data with normal distribution before and after the operation was performed by paired *t*-test. Data with non-normal distribution were compared by the Willcoxon test. The paired comparison of preoperative data, and three months and one year after operation were compared by analysis of variances (ANOVA) of repeated measurements or Friedman test. All of the statistical analyses were two sides, and *p* < 0.05 was considered statistically significant.

## 3. Results

A total of nine patients (eight males and one female) were included in this study between April 2019 and September 2019. Six patients underwent left Vim ablation, and three patients underwent right Vim ablation. The baseline characteristics of the nine patients are shown in Table 1.

The parameters in the MRgFUS surgery were as follows: the mean ablation time of the 9 patients was 6.22 ± 2.64 s, and the average maximum temperature was 57.89 ± 1.36 °C. Figure 1 shows the MRI images of the patient No. 3 before, during, and at one month, three months, and one year after the operation.

The tremor of contralateral limbs in all patients improved significantly after Vim ablation by MRgFUS. Table 2 and Figure 2 shows the scores of MDS-UPDRSIII items at different follow-up times. In the OFF state, the total tremor score, tremor score of contralateral/ipsilateral limbs, total bradykinesia score, bradykinesia score of contralateral/ipsilateral limbs, and rigidity score of contralateral limbs all significantly improved at three months and one year after the operation, compared with the preoperative scores (all *p* < 0.05), indicating that unilateral Vim ablation by MRgFUS could improve the long-term tremor, bradykinesia, and rigidity of contralateral limbs, as well as tremor and bradykinesia of ipsilateral limbs, and the short-term rigidity of ipsilateral limbs (within three months). In the ON state, the total tremor score and tremor score of the contralateral side significantly improved at three months and one year after operation (all *p* < 0.05) compared with the scores before operation. In addition, the scores in the ON state were significantly lower than in the OFF state, indicating that a combination of drug therapy with unilateral Vim ablation by MRgFUS could further improve the tremor of contralateral limbs in TDPD patients for a long time. The scores of other items in MDS-UPDRSIII at three months and one year after operation were not significantly different from the scores before the operation, except for the total rigidity score and rigidity score of the ipsilateral side at one year after the operation, which was increased (*p* < 0.05). These findings indicated that the ipsilateral rigidity progressed at one year after operation in the ON state than pre-operation, while the contralateral rigidity did not significantly change compared to pre-operation.

Tremor in nine patients was also assessed by CRST score. Figure 3A shows the total CRST score before and at three months and one year after operation in ON and OFF states. Figure 3B shows the CRST (A + B) score of contralateral limbs before and at three months and one year after operation in ON and OFF states. Friedman test showed that compared with baseline, in the OFF state, the total CRST score significantly improved at three months (*p* = 0.007) and one year (*p* = 0.029) after the operation, but the difference between three months and one year after the operation did not significantly differ (*p* = 1.000). Similarly, in the ON state, the total CRST score significantly improved at three months (*p* = 0.007) and one year (*p* = 0.010) after the operation compared with baseline, and there was no difference between three months and one year after the operation (*p* = 1.000). These findings demonstrated that the total tremor score in both ON and OFF states significantly improved at three months and one year after unilateral Vim ablation by MRgFUS.

In the OFF state, the CRST (A + B) score of contralateral limbs improved at three months (*p* = 0.014) and one year (*p* = 0.004) after the operation compared with before operation. However, the differences between three months and one year after the operation did not significantly differ (*p* = 1.000). Similarly, in the ON state, the CRST (A + B) score of contralateral limbs improved at three months (*p* = 0.040) and one year (*p* = 0.001) after operation compared with before operation and there was no difference between three months and one year after the operation (*p* = 0.867). These findings demonstrated that the tremor score of contralateral limbs in both ON and OFF states significantly improved at three months and one year after unilateral Vim ablation by MRgFUS.

The median LEDD before the operation, and at one and three months, as well as one year after operation was 600 (456, 600) mg/d, 475 (337, 600) mg/d, 475 (387, 600) mg/d, and 575 (425, 612) mg/d, respectively, which did not significantly differ from Friedman test. The medication of nine patients before and after the operation is listed in Appendix A in detail. These findings demonstrated that the LEDD was not significantly reduced at one and three months, as well as one year after unilateral Vim ablation by MRgFUS. In addition, the CRST scores at baseline, three months and one year after operation in the ON state were significantly lower than at OFF state (*p* < 0.05) (Figure 3A,B), indicating that the drug therapy still has an important role in alleviating tremor in TDPD patients.

Both intraoperative and postoperative adverse responses were found in the patients. The intraoperative adverse responses mainly included headache (*n* = 1) and dizziness (*n* = 2), which disappeared after the operation was completed. The adverse responses on the day after the operation mainly included gait disturbance (*n* = 3), tongue tip numbness (*n* = 4), and hypogeusia (*n* = 1). The gait disturbance in two patients and tongue tip numbness in one patient disappeared one month after the operation. All other adverse responses in patients improved from 3–12 months after the operation, and patients reported that their daily living was not substantially influenced. All of the adverse responses were mild to moderate, and no patient received glucocorticoids treatment.

## 4. Discussion

This small sample clinical study provides a preliminary discussion of the effectiveness and safety of unilateral MRgFUS Vim ablation in TDPD patients through one year of follow-up. Our primary outcome is that treating TDPD with unilateral Vim ablation by MRgFUS can improve the symptoms of limb tremor as well as some non-motor symptoms without serious adverse reactions. The second outcome is that in the OFF state we also observed improvements in other motor functions involving the contralateral and ipsilateral limbs, which is a new discovery compared with previous studies.

### 4.1. Features of the Study

Findings from the nine consecutive TDPD patients who underwent unilateral Vim ablation by MRgFUS showed that the contralateral tremor significantly improved after the treatment. The improvement of symptoms after MRgFUS could be due to surgical benefits, drug effects, and placebo effects. Surgical benefits refer to the treatment effects acquired directly from MRgFUS ablation. These short-term benefits could be determined during the operation. For instance, the alleviation or disappearance of tremor during operation could be considered the effect of surgery. For both doctors and patients, the more important issue is whether the substantial tremor reduction or disappearance is a long-term or short-term improvement caused by unilateral Vim by MRgFUS. Nevertheless, many patients often need to be combined with drug adjuvant therapy after the operation. In addition, tremor and other PD-related motor symptoms are substantially influenced by patients’ subjective feelings. Therefore, the postoperative improvement of motor score can mainly be interpreted as the joint effect of surgical benefits, drug therapy, and placebo effects. Most previous studies used the change in CRST score in the ON state as the indicator for assessing the treatment effects. However, as the doses of drugs used for assistant therapy could increase or decrease at different follow-up times, we speculated that using the CRST score at ON state is not very rigorous, even for TDPD patients with drug resistance before the operation. To further investigate the long-term efficacy of unilateral Vim ablation by MRgFUS in TDPD patients, we used the MDS-UPDRSIII score and CRST score in the OFF state to rule out the effects of the drug therapy. In addition, some randomized controlled trials have also reported CRST score improvement within three months in the control group, and such placebo effects generally disappeared three months after operation [6]. Therefore, three months and one year after the operation, which was selected as time points for follow-up, could generally rule out the placebo effects.

### 4.2. Improvement of Tremor

Our findings showed that in the OFF state, the tremor score at three months and one year after the operation improved compared to baseline. Specifically, compared with the preoperative score, the median MDS-UPDRSIII tremor score (total score of items No. 3.15–3.18) of contralateral limbs was reduced by 50% (4.0 vs. 8.0) and 62.5% (3.0 vs. 8.0), and median CRST (A + B) score of contralateral limbs was reduced by 80% (4.0 vs. 20.0) and 85% (3.0 vs. 20.0) at three months and one year after the operation, respectively. However, the Friedman test showed no significant difference between three months and one year after the operation. These findings demonstrated that unilateral Vim ablation by MRgFUS could significantly improve the tremor in TDPD patients and that the treatment effects could be maintained for about one year. The scores were also compared in the ON state, revealing similar results. Moreover, the MDS-UPDRSIII score and CRST score in the ON state were both lower compared to the OFF state, indicating that unilateral Vim ablation by MRgFUS in combination with drug therapy could further improve the manifestations of patients and indicated that patients still had good responses to drug therapy within one year after the operation. Notably, in contrast to the other studies that included drug-resistant TDPD patients, the TDPD patients included in this study also had good responses to drug therapy at baseline. Therefore, the findings of this study may have higher guiding significance for patients unsatisfied with the effects of drug therapy or with relatively high requirements for motor functions.

### 4.3. Improvement of Other Motor Functions

Some case reports also revealed that patients acquired other motor improvements in addition to tremor alleviation. However, these findings need to be further verified. The findings in this study demonstrated that the OFF state scores of bradykinesia and rigidity of contralateral limbs in MDS-UPDRSIII at three months and one year after operation significantly improved compared to that before the operation, but no significant difference was found in the ON state. We speculated that this might be due to the following reasons: (1) unilateral Vim ablation by MRgFUS could alleviate the contralateral bradykinesia and rigidity to a certain extent. Previous studies mainly used the scores in ON state as the indicator, in which drug effects could mask the treatment effects of surgery, and thus the findings could appear statistically insignificant. Our findings in ON state verified this hypothesis from another perspective; (2) in the OFF state, contralateral rigidity improved significantly at three months and one year after operation, while the ipsilateral side exhibited no obvious changes. Notably, in the On state, ipsilateral rigidity deteriorated at one year after operation (LEDD did not significantly change), but the contralateral side remained stable, thus demonstrating that unilateral MRgFUS Vim ablation could presumably delay the progression of contralateral rigidity (such as offset the decline of drug treatment effect) from another aspect; and (3) patients with some subtypes of TDPD might acquire higher motor benefits from unilateral Vim ablation. However, more studies are needed to explore the characteristics of such patients. In addition, we also tested the hypothesis that ipsilateral limbs could also benefit from the treatment. The findings of this study demonstrated that the OFF state MDS-UPDRSIII tremor score of ipsilateral limbs at three months and one year after operation significantly improved than before the operation, but in the On state, the scores did not significantly differ. These findings indicated that unilateral Vim ablation by MRgFUS could improve the tremor of bilateral limbs, which could be associated with the electrophysiological pathways of tremor. Our previous study using structural connectivity map identified a subnetwork sensitive to MRgFUS thalamotomy with a U-shaped tendency that was critical for clinical tremor recovery, which was further confirmed significantly correlated with D1 and D2 receptors, DAT, and F-DOPA. The previous results help to understand the mechanism of MRgFUS Vim ablation improving tremor from a structural perspective, yet are not enough to explain all of the findings in this study [7]. However, more studies are needed to investigate the mechanisms.

### 4.4. Doses of Drugs

The median LEDD was not significantly reduced after operation in nine patients than before the operation. However, the LEDD in four patients did reduce one year after the operation than before the operation (14.8–33.3%), while it remained unchanged or slightly increased in the other patients. The findings of LEDD in this study were consistent with the results reported by Yamamoto et al., showing that only some patients were with LEDD reduction, while the drug dose remained unchanged or even increased in the other patients to meet the requirements for motor functions.

### 4.5. Adverse Responses

Anatomically, nuclei close to Vim include ventral oralis anterior nucleus (Voa, also known as ventral anterior nucleus [VA]), ventral oralis posterior nucleus (Vop, also known as ventral lateral nucleus [VL]), ventral caudalis nucleus (VC, also known as ventral posterior nucleus [VP]) and posterior margin of the internal capsule (IC). One small difference between VP and VC is that VP consist of Vim, ventral posterolateral nucleus (VPL), and ventral posteromedial nucleus (VPM), while VC is composed of VPL and VPM. Voa and Vop participate in the regulation of somatic movements, and Vim + Vop have also been used as targets of MRgFUS treatment. VPL is associated with the depth sensation of the body and limbs, and VPM is associated with the depth sensation of the head and facial areas. Adverse events related to Vim ablation by MRgFUS are associated with injuries of structures around targets. However, the nuclei around Vim still lack apparent contrasts with surrounding structures even on high-resolution MRI, and the positioning of Vim can only rely on indirect methods or preoperative diffusion tensor imaging (DTI), while the indirectly calculated ablation volume is only about 0.5−2% of the Vim volume [8]. Most adverse events are mild to moderate and associated with edema around targets. Such adverse events are generally reversible and could disappear after a recession of edema (within three months). Severe adverse events such as limb paralysis and limb numbness are associated with midline-lateral injuries of IC, which in some patients can be improved by steroid pulse therapy. However, some other patients may live with sequelae for a long time. In this study, one patient had headaches and two patients have had dizziness during the operation, all of which disappeared after the operation. The postoperative adverse reactions mainly included gait disturbance (*n* = 3), tongue tip numbness (*n* = 4), and hypogeusia (*n* = 1). The gait disturbance in two patients, as well as tongue tip numbness in one patient, disappeared one month after the operation. All of the adverse events further improved in the following year of follow-up, having no substantial influence on the daily living of patients. Compared with the findings reported by Yamamoto and Sinai et al., we found fewer intraoperative adverse reactions in our study [9,10]. The postoperative adverse reactions mainly included gait disturbance and sensory disturbance, while no motor disturbance of limb was found. Although the adverse responses in some patients did not disappear within the year of follow-up, the symptoms substantially improved and did not influence their daily livings. The above adverse events were still associated with injuries of VPL and VPM structures, indicating that the anterior-posterior directional ablation of AC-PC should be further controlled.

### 4.6. Recurrence

Several previous studies have reported tremor recurrence at one year or longer follow-up. For instance, Yamamoto et al. reported that one patient (9%) experienced tremor recurrence within one year of follow-up. Sinai et al. reported that one patient (4%) experienced recurrence at five years after the operation. Currently, the basic consensus on recurrence is the insufficient effective ablation range, but the possibility of rapid progression of the disease after MRgFUS should not be ignored [9,10]. In this study, no evident tremor recurrence occurred within the one-year follow-up in the nine patients, which could be associated with the higher average ablation volume than in cases reported in other studies. These findings demonstrated that surgery design and control of intraoperative parameters, such as ablation times and ablation temperature, could have important influences on long-term prognoses. Operators require a certain learning curve for improving the skillfulness of MRgFUS ablation. Our center is the first that introduce the MRgFUS technique in China, which was used to complete treatments in 41 ET patients and 10 TDPD patients up to November 2021, revealing certain advancements in the treatment experience. In addition, we also explored additional targets of treatment in patients with different types of PD. A prospective study on TTP ablation by MRgFUS for the treatment of PD is currently ongoing.

## 5. Limitations

There are several limitations to this study. Firstly, the sample size was relatively small. MRgFUS technique was introduced to our hospital in 2019, thus making our medical center among the first to use this technique. However, the application of MRgFUS in neurosurgery is still in the initial stage, and thus only a few TDPD patients completed long-term follow-up in the out center. Secondly, Vim was still positioned with the indirect method in this study, while more individualized positioning methods, such as dental nucleus- locus ruber-thalamic fasciculus imaging and real-time prediction of ablation area, could further reduce the operation error. In fact, this is one of the directions for further improvement of the operation quality and reduction of adverse responses.

## 6. Conclusions

Unilateral Vim ablation by MRgFUS for the treatment of TDPD significantly improved the tremor of contralateral limbs. Certain degrees of improvements in a tremor of bilateral limbs, bradykinesia, and control rigidity were found in patients in the OFF state, and the efficacy was generally maintained stable within one year. The motor score was lower in the ON state than in the OFF state, indicating that a combination of drug therapy could better exert the treatment effects in TDPD patients. In summary, treating TDPD with unilateral Vim ablation by MRgFUS could not only improve the symptoms of limb tremor but could also improve the other core symptoms, such as bradykinesia and rigidity, as well as some non-motor symptoms, and the symptoms of ipsilateral limbs. More studies are needed to further clarify the Vim-related fiber conduction structures, improve the surgery design, and help TDPD patients to benefit more from the treatment.

## Figures and Tables

**Figure 1 jcm-11-04502-f001:**
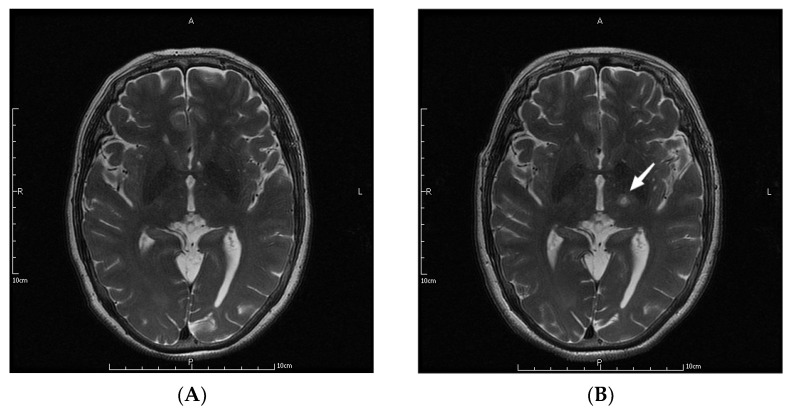
MRI images of a patient before (**A**), during (**B**), and at one month (**C**), three months (**D**), and one year (**E**) after the operation. The letters in the pictures are abbreviations of anterior (A), posterior (P), left (L) and right (R). The white arrow pointed to the lesions created by MRgFUS Vim ablation.

**Figure 2 jcm-11-04502-f002:**
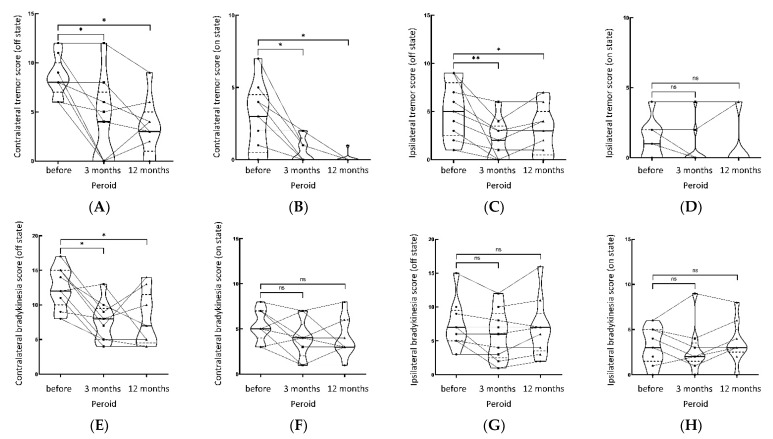
Shows the tremor, bradykinesia, rigidity part of MDS-UPDRS score (on/off state) of contralateral and ipsilateral limbs pre-operation, 3 months, and 12 months post-operation, respectively. (**A**) Contralateral tremor score (off state), (**B**) contralateral tremor score (on state), (**C**) ipsilateral tremor score (off state), (**D**) ipsilateral tremor score (on state). (**E**) Contralateral bradykinesia score (off state), (**F**) contralateral bradykinesia score (on state), (**G**) ipsilateral bradykinesia score (off state), (**H**) ipsilateral bradykinesia score (on state). (**I**) Contralateral rigidity score (off state), (**J**) contralateral rigidity score (on state), (**K**) ipsilateral rigidity score (off state), (**L**) ipsilateral rigidity score (on state). ** *p*< 0.01, * *p* < 0.05, NS not significant.

**Figure 3 jcm-11-04502-f003:**
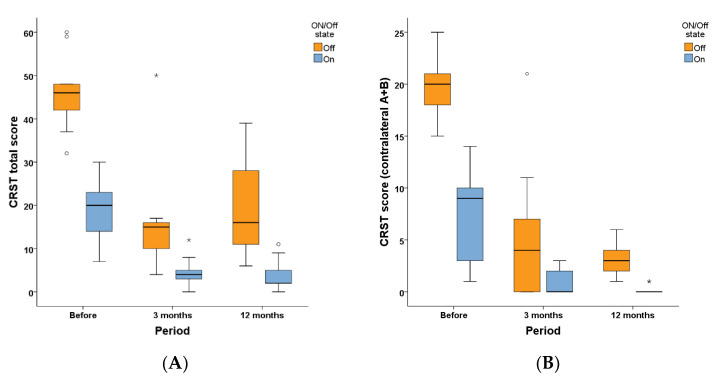
CRST scores before and after MRgFUS Vim ablation. (**A**) Total CRST score at ON and OFF state before and at three months and one year after unilateral Vim ablation by MRgFUS. (**B**) CRST (A + B) score of the contralateral limb at ON and OFF state before and at three months and one year after unilateral Vim ablation by MRgFUS. ◦ and * in the figures represent the outliers of the corresponding group. MDS-UPDRSIII: Movement Disorder Society Unified Parkinson’s Disease Rating Scale part III; CRST: Clinical Rating Scale for Tremor.

**Table 1 jcm-11-04502-t001:** Baseline characteristics of the patients.

Characteristics	Number of Patients (*n* = 9)
Age (years)	64.7 ± 6.1
Gender (Male, %)	8 (88.9%)
Disease duration (years)	7.0 (5.5, 9.0)
MDS-UPDRSIII (OFF state)	
MDS-UPDRSIII (C)	26.0 (22.0, 27.5)
MDS-UPDRSIII (I)	15.0 (11.0, 18.5)
Total MDS-UPDRSIII score	55.0 (48.0, 65.5)
MDS-UPDRSIII (OFF state)	
MDS-UPDRSIII (C)	11.0 (9.0, 13.5)
MDS-UPDRSIII (I)	7.0 (3.0, 8.0)
Total MDS-UPDRSIII score	26.0 (17.0, 27.0)
CRST (OFF state)	
CRST A + CRST B (C)	20.0 (17.5, 21.0)
CRST A + CRST B (I)	10.0 (7.0, 17.0)
Total CRST score	46.0 (39.5, 53.5)
CRST (ON state)	
CRST A + CRST B (C)	9.0 (3.0, 10.5)
CRST A + CRST B (I)	1.0 (0.0, 6.0)
Total CRST score	20.0 (13.0, 23.5)
MMSE	26.0 (23.5, 27.0)
Barthel index	100 (85, 100)
PDQ-39	24.0 (17.5, 41.0)
Hoehn and Yahr stage	2.0 (2.0, 3.0)
LEDD (mg/d)	600 (456, 600)
SDR	0.48 (0.42, 0.62)
NMS	22.0 (9.5, 24.5)

Age was expressed as mean (standard deviation), and others as median (interquartile range). Abbreviations: (C) contralateral side; (I) ipsilateral side; MMSE, Mini-mental State Examination; PDQ-39, Parkinson’s Disease Questionnaire for daily living; LEDD, levodopa equivalent dose; SDR, skull density ratio; NMS, non-motor symptom score. MDS-UPDRSIII: Movement Disorder Society Unified Parkinson’s Disease Rating Scale part III; CRST: Clinical Rating Scale for Tremor.

**Table 2 jcm-11-04502-t002:** Comparison of MDS-UPDRSIII scores during operation with before operation.

MDS-UPDRSIII (No. 3.1–3.18)	OFF State			ON State		
	Before	3 Months	1 Year	Before	3 Months	1 Year
Speech	1.0 (1.0, 2.0)	1.0 (1.0, 2.0)	1.0 (1.0, 2.0)	1.0 (0.0, 1.0)	1.0 (0.5, 1.0)	1.0 (1.0, 1.0)
Facial expression	2.0 (1.0, 2.5)	2.0 (1.0, 2.5)	2.0 (1.5, 3.0)	1.0 (1.0, 3.0)	1.0 (1.0, 2.0)	2.0 (1.5, 2.0)
Tremor	19.0 (14.5, 21.0)	8.0 (5.0, 10.5) **	7.0 (4.0, 12.5) **	6.0 (1.5, 11.0)	2.0 (0.0, 2.5) *	0.0 (0.0, 2.5) *
Contralateral	8.0 (7.0, 10.0)	4.0 (0.0, 7.0) *	3.0 (0.5, 4.5) *	3.0 (0.5, 4.5)	0.0 (0.0, 1.5) *	0.0 (0.0, 0.0) *
Ipsilateral	5.0 (2.5, 8.0)	2.0 (0.0, 3.5) **	3.0 (0.5, 5.0) *	1.0 (0.0, 2.0)	0.0 (0.0, 1.0)	0.0 (0.0, 2.0)
Bradykinesia	23.0 (16.5, 25.0)	16.0 (9.5, 19.5) *	17.0 (10.0, 23.5) *	8.0 (6.5, 12.0)	6.0 (4.5, 10.5)	9.0 (5.5, 10.5)
Contralateral	12.0 (10.0, 15.0)	8.0 (5.0, 9.5) *	7.0 (4.5, 11.5) *	5.0 (4.0, 7.0)	4.0 (2.0, 5.5)	3.0 (3.0, 6.0)
Ipsilateral	7.0 (5.0, 9.5)	6.0 (2.5, 9.0) *	7.0 (3.5, 9.0)	3.0 (1.5, 5.0)	2.0 (1.5, 3.5)	3.0 (2.5, 5.0)
Rigidity	9.0 (7.0, 11.0)	8.0 (7.0, 11.0)	9.0 (7.0, 11.5)	6.0 (5.5, 6.5)	6.0 (5.5, 8.5)	7.0 (6.0, 10.0) *
Contralateral	3.0 (2.5, 4.5)	3.0 (2.0, 3.5) *	3.0 (2.0, 4.0) *	2.0 (2.0, 3.0)	2.0 (2.0, 3.0)	2.0 (2.0, 3.5)
Ipsilateral	3.0 (2.0, 4.5)	4.0 (2.0, 4.0)	3.0 (2.0, 4.0)	2.0 (1.5, 2.5)	2.0 (2.0, 4.0)	3.0 (2.0, 4.0) *
Stand	1.0 (0.0, 1.0)	1.0 (0.0, 1.0)	1.0 (0.0, 1.0)	0.0 (0.0, 0.0)	0.0 (0.0, 0.5)	0.0 (0.0, 0.0)
Posture	1.0 (1.0, 2.0)	2.0 (1.0, 2.0)	2.0 (1.0, 3.0)	1.0 (0.0, 1.0)	1.0 (0.5, 2.0)	1.0 (1.0, 2.0)
Gait	1.0 (1.0, 2.0)	1.0 (1.0, 2.0)	1.0 (1.0, 2.0)	1.0 (0.0, 1.0)	1.0 (0.5, 1.0)	1.0 (0.0, 1.0)
Freeze	0.0 (0.0, 1.0)	0.0 (0.0, 0.5)	0.0 (0.0, 1.5)	0.0 (0.0, 0.0)	0.0 (0.0, 0.0)	0.0 (0.0, 0.0)
Postural stability	1.0 (0.0, 2.5)	0.0 (0.0, 3.0)	0.0 (0.0, 3.0)	0.0 (0.0, 0.0)	0.0 (0.0, 0.0)	0.0 (0.0, 1.0)
Total score	55.0 (48.0, 65.5)	45.0 (26.5, 50.0) *	40.0 (30.0, 60.5) **	26.0 (17.0, 27.0)	18.0 (14.0, 25.5)	21.0 (17.5, 27.0)

* *p* < 0.05, ** *p* < 0.01, comparison of scores at three months and one year after the operation with baseline. Tremor score was the total score of questions No. 3.15–3.18; bradykinesia score was the total score of questions No. 3.4–3.8 and 3.14; rigidity score was the score of question No. 3.3. Contralateral referred to the contralateral limbs of Vim ablation, and ipsilateral referred to the ipsilateral limbs of Vim ablation. MDS-UPDRSIII: Movement Disorder Society Unified Parkinson’s Disease Rating Scale part III.

## Data Availability

The datasets used and/or analysed during the current study are available from the corresponding author on reasonable request.

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
