# Peer review of "Comparison of Motor Scores between OFF and ON States in Tremor-Dominant Parkinson’s Disease after MRgFUS Treatment"

_jcm, 2022, doi:10.3390/jcm11154502_

Round 1

Reviewer 1 Report

This manuscript reported a small,  none placebo-controlled, unilateral VIM ablation study using MRgFUS in TDPD patients. The authors reported surgical outcomes by subjective assessments: MDS-UPDRSIII and CRST at three months and one-year post-operation.  The authors found significant improvements in the tremor of contralateral limbs and some improvement in other core PD symptoms. The research team compared the assessment in ON and OFF states to unmask the surgical benefit from the drug effect. They also reported no significant drug intake reduction and no serious adverse responses. 

This is a well-organized, clearly presented clinical report, parallel with Gallay's in Switzerland, Sinai's in Isreal, and Yamamoto's in Japan. This study mirrored the subjective assessment in the literature mentioned above, thus streamlining the cross-comparison in the future.  This manuscript adds valuable information for neurosurgeons to further refine and standardize the MRgFUS technique in China and beyond. 

However, there are a few points for improvement:

1. In line 104, Exclusion Criteria #5, the authors need to define what is "with contradiction to MRI." 

2. In lines 108-109, Disease Condition Assessment, the authors need to stress that although the scores were assessed with at least two neurological physicians, the evaluation (both ON/OFF state and contralateral/ipsilateral side) was not blind or randomized.  

3. In Figure 1, the volume and orientation of VIM ablation were not apparent, even though the authors emphasized the importance of the orientation in lines 324-326 and the volume in lines 333-335. 

4. In lines 202-208, the authors discuss the none significant drug intake reduction with only LEDD comparison.  It could be helpful to list all the drug therapy for the nine patients, especially anticholinergic agents. Patients might benefit from the treatment by substituting a more tolerable yet previously less effective drug post-surgery without lowering the LEDD. 

5. In lines 275-280, Improvement of other Motor Functions, the authors discussed the improvement of contralateral rigidity in ON and OFF states compared with the ipsilateral side. The authors need to clarify the second reason and shorten the sentence. 

6. In lines 287-288, the authors did not cite their early work in NeuroImage 2021 regarding the mechanisms in MRgFUS (one of very few with mechanistic insight). 

7. In lines 320, 329, and 330, the authors omitted reference numbers [8] and [9] in the text.

8. The authors might want to add Gally's 51 patients cohort (Fronters in Surgery 2020) into the comparison with Yamamoto's [8] and Sinai's [9]. 

9. Suggestion: the authors might want to consider using objective assessments, such as TekScan MatScan for postural sway or voice analysis for tremor in company with subjective assessments.  

Reviewer 2 Report

As the authors mentioned, this study described the first MRgFUS application in China, which is very encouraging for the PD patients in China who cannot benefit from pharmacological treatment. It verified that unilateral Vim ablation by MRgFUS could significantly improve the tremor of contralateral limbs in TDPD patients. Moreover, it also found that certain motor function got improved after MRgFUS. These improvements could last up to one year. I only have a few minor comments.

Minor comments:

11)      Please check with JCM if you should define all the abbreviations, such as TDPD, MRgFUS, CRST, LEDD et al. in the Abstract.

22)      Please add scale bar in Figure 1 and point to the target with arrows.

3)      In Table 1, are the numbers in the parentheses 25% and 75% percentage or standard deviation? Please clarify.

43)      Please use larger fonts in Figures 2 and 3.

54)      It mentioned that some patients still took drugs after the MRgFUS. Can the authors provide more details such as frequency and dosage since the drug could partially contribute to the improvement of tremor.

Reviewer 3 Report

In this study, the authors prospectively followed up the patients for 1 year and investigated the motor function, adverse reactions, and adjustment of drug doses in ON and OFF states in TDPD patients after the unilateral ventral intermediate nucleus of thalamus (Vim) ablation by MRgFUS. They conclude that Treating TDPD with unilateral Vim ablation by MRgFUS could improve the symptoms of limb tremor and the other core symptoms, such as bradykinesia and rigidity, as well as some non-motor symptoms and the symptoms of ipsilateral limbs. However, there are important issues that must be addressed.

·         The sample size is small, the authors must validate the results. If two/three/four replications were done, then samples should be pooled and analyzed as such. The sample size should reflect what was used in the statistical analyses.

·         Why do the authors include 8 males and 1 female? Are there any gender differences? The authors should have included an equal number of males and females.

·         The Discussion should summarize the findings of the study and connect the different results.

·         What are your significant values?

Round 2

Reviewer 3 Report

The authors have addressed all my comments.

Author Response

We deeply appreciate your suggestions on the revision of this article.